# Incrementality Bidding via Reinforcement Learning under Mixed and Delayed Rewards

**Ashwinkumar Badanidiyuru**
Google
ashwinkumarbv@google.com

**Zhe Feng**
Google Research
zhef@google.com

**Tianxi Li**
University of Virginia
tianxili@virginia.edu

**Haifeng Xu**
University of Chicago
haifengxu@uchicago.edu

## Abstract

Incrementality, which measures the causal effect of showing an ad to a potential customer (e.g. a user in an internet platform) versus not, is a central object for advertisers in online advertising platforms. This paper investigates the problem of how an advertiser can learn to optimize the bidding sequence in an online manner *without* knowing the incrementality parameters in advance. We formulate the offline version of this problem as a specially structured episodic Markov Decision Process (MDP) and then, for its online learning counterpart, propose a novel reinforcement learning (RL) algorithm with regret at most $\widetilde{O}(H^2\sqrt{T})$, which depends on the number of rounds $H$ and number of episodes $T$, but does not depend on the number of actions (i.e., possible bids). A fundamental difference between our learning problem from standard RL problems is that the realized reward feedback from conversion incrementality is *mixed* and *delayed*. To handle this difficulty we propose and analyze a novel pairwise moment-matching algorithm to learn the conversion incrementality, which we believe is of independent interest.

## 1  Introduction

Nowadays, online advertising systems (e.g. Google, Facebook, and Amazon) have demonstrated their significant power for connecting advertisers and potential customers. Moreover, automated auctions are widely used in online ad platforms for matching advertisers and users, and for price discovery [23, 8]. Many advertisers are bidding repeatedly online to show ads to users. Therefore, an important problem for advertisers is to design efficient bidding algorithms to maximize their accumulated utility. Moreover, this is also an important problem for online ad platforms, because *auto-bidding* — i.e., the advertisers authorize ad platforms to bid on behalf of them —has become prevalent in online advertising [9, 11].

A growing body of work is investigating the problem of *learning to bid* in repeated auctions [25, 10, 5, 13, 12, 4, 20, 19]. However, existing work assumes that the value of showing ads to a user (after winning the auction) comes from either an oblivious adversary or is sampled from an underlying fixed distribution. Unfortunately, this ignored the crucial *causal effect* of showing an ad to a potential customer and cannot capture how much an ad will *change* the user's conversion rate, which should have been what the advertiser is bidding for. For example, if a user sees the same ad multiple times in a very short time period, then the conversion [1] of this user may not be much better — in fact, may

---

[1] A conversion represents a desired interaction between the user and advertiser, e.g., download the App or buy the products from the advertiser.

even be worse — than that of showing ads to this user only once within this period. Such multiple redundant ad placements are commonly known in reality to be cost-inefficient for the advertisers, but cannot be captured by previous models of learning to bid with either random or completely adversarial ad values. To overcome the drawbacks of previous approaches in measuring the value of an advertising opportunity, a recent visionary whitepaper by Lewis and Wong [18] introduced the important notion of *incrementality*, which properly quantifies this causal effect of showing an ad to the user as validated by massive real data from leading advertising platforms. However, [18] focuses on the offline learning problem of estimating incrementality from past data. The online learning of incrementality parameters, as well as how to utilize it to optimize bidding sequences, has not been studied; this is what we embark on in this paper.

In Figure 1, we visualize the conversion incrementality caused by three ad placements and used the colored areas to capture the incrementality caused by each ad placement. There are two main challenges in learning incrementality, i.e., *delayed* and *mixed* conversion feedback. First, the conversion may not happen immediately when the ad is shown to the user, even though the conversation

rate may quickly peak after showing the ad (see Ad 2 and Ad 3 in Figure 1). Such delayed conversion feedback in online advertising is studied only recently in offline setups, e.g., [6, 21, 22, 3]. Second, when a conversion happens, it is a mixed effect from multiple previous ad placements before the conversion. In this work, we formulate incrementality bidding as an episodic Markov Decision Process (MDP), where each episode represents an interaction with a single user drawn from a population of the same characteristics (and thus assumed to have the same incrementality parameters). The advertiser wants to learn the incrementality and optimize bidding for this user population. This is hardly possible in general given the two difficulties above as well as the potentially complex dependence of incrementality on the entire history. We thus make a Markovian assumption that the conversion incrementality of an ad placement at each round $h$ only depends on its *last* ad impression shown before this round. In addition, we introduce the *heterogeneous Poisson process* to formally capture the mixed and delayed rewards feedback in the conversion process (Poisson process is also mentioned by Lewis and Wong [18], but only qualitatively).

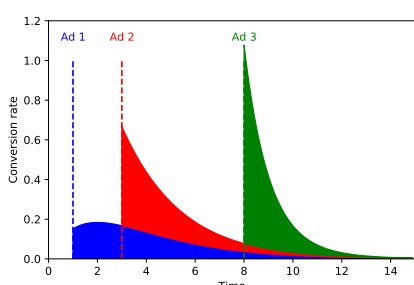

Figure 1: Toy example: visualization of the conversion incrementality triggered by three ad impressions. The rate of conversion incrementality of each ad follows Gamma(2, 2), Exp(0.5) (shifted right by 3) and Exp(1) (shifted right by 8) distributions, respectively.

To optimize the sequential strategy for incrementality bidding in online learning, there is an intrinsic exploration versus exploitation tradeoff. On one hand, the advertiser needs to bid high in order to win the ad placement opportunity and also to learn the conversion incrementality. On the other hand, she also doesn't want to bid too high such that her conversion reward may be less than her cost, i.e., the payment due to winning the auction. This observation naturally motivates our design of a model-based RL algorithm that adopts a UCB-style approach to balance exploration and exploitation.[2]

**Our Results and Contributions.** The main contributions of this paper are two folds: (1) we introduce an episodic MDP, coupled with a heterogeneous Poisson process, to capture the incrementality bidding problem; (2) we design a novel reinforcement learning algorithm for the advertiser to optimize her sequential bidding while learning incrementality simultaneously. *Technically*, our main result is the design of an RL algorithm for incrementality bidding that provably has regret at most $\widetilde{O}(H^2\sqrt{T})$ and is near-optimal in terms of the dependence on $T$. *Conceptually*, our result demonstrates the possibility of designing highly-efficient RL algorithms for incrementality bidding despite its various challenges. To the best of our knowledge, this is the first work that analyzes incrementality bidding in an online setting theoretically.

Notably, our regret bound is independent of the number of possible bids (equiv. actions in standard RL). This is due to the special structures of the incrementality bidding problem, which has a small *effective* action space. Specifically, there are only two effective outcomes for an advertiser—winning

---

[2]We remark that designing model-free no-regret RL algorithms is an intriguing open problem. Due to delayed and mixed reward feedback, it appears challenging to use algorithms like Q-learning in the incrementality bidding problem.

or losing— that eventually affect the state transitions. The conversion incrementality only happens if the learner wins. Finally, we remark that our results are independent of auction formats and all our results hold for second price auctions, first price auctions, and other standard single-item auction formats. To achieve this regret bound, we provide a novel regret decomposition to incorporate with the convergence rate analysis of our novel parameter estimation method.

One key technical novelty of this paper is a new parameter estimation method, *Pairwise Moment-Matching* (`PAMM`) algorithm, that can estimate reward parameters under mixed and delayed conversion feedback. In this type of models with mixture effects, standard likelihood-based model fitting procedure would lead to solving non-convex optimization problems resulting in difficulties for computation and analysis, as well as unpleasant gap between theory and algorithm [15]. However, `PAMM` estimates incrementality parameters in an online and computationally efficient manner without the need of solving nonconvex optimization problems. The estimators are provably consistent with a nearly-optimal convergence rate. We believe this novel technique will be of independent interest for other online learning problems with delayed and mixed reward feedback.

**Related Work.**  As mentioned in the introduction, our work is highly inspired by the visionary work of Lewis and Wong [18], who firstly proposed incrementality bidding (and attribution). However, the method proposed in [18] lacks a theoretical guarantee and cannot be adapted to online learning settings. Our work is generally related to the *learning to bid* literature [25, 10, 5, 13, 12, 4]. However, all these existing papers usually model this problem as (contextual) bandits and don't consider the causal effect of showing an ad to the user captured by incrementality (i.e., the additional value or conversion given the ad placements in previous rounds), which causes the main difference from this paper. Specifically, in all these previous works [25, 10, 5, 13, 12, 4], the reward for an ad impression is realized immediately at the current round whereas, in our model, any ad impression only increases the rate of conversions, which however may be realized in any later time following a Poisson process and, more importantly, is mixed with the incrementality of other ad impressions. We believe our model captures the effect of advertising more realistically. Some other less related works include delayed feedback in other different RL problems, which have only been studied recently [17, 14]. We handle delayed conversion feedback through a parametric inhomogeneous Poisson process, which is different from these works.

## 2   The Sequential Optimization Problem of Incrementality Bidding (`IB`)

This paper adopts the perspective of an online advertiser, and studies the advertiser's learning problem of optimizing her sequential bidding policy. Next, we will first describe the background and motivation of this problem and then introduce the formal model.

We consider the sequential bidding problem for a single advertiser — i.e., the *decision maker* or *learner* — in online advertising systems. The advertiser participates online advertising auctions in order to win opportunities for showing ads to Internet users and, ultimately, gain conversions (i.e., purchases of products or services) from Internet users. The interaction between the advertiser and Internet users happens in an *episodic* manner; each episode corresponds to one Internet user. Specifically, each episode has $H$ rounds, during which the advertiser (learner) will interact with the user of that episode for at most $H$ times. Notably, $H$ is a finite number since (1) an interaction happens only when a user visits a particular webpage and (2) if an advertiser already bids for a user for sufficiently many times and still was not able to get conversion, the advertiser typically will cease to advertise to this user (at least for some time). We use $T$ to denote the total number of episodes.[3] To focus on the fundamentals of the problem, this work studies the basic setup with $T$ i.i.d. episodes (i.e., Internet users) that occur *sequentially* and evenly distributed $H$ rounds at integer time point $1, 2, \cdots, H$ within each episode. Interesting future directions include examining the more general settings in which each Internet user in the corresponding episode may depend on a context feature, the time points of the $H$ rounds are different for different episodes, and the episodes may happen in batch (i.e., multiple similar users arrive at the same time).

**Conversion incrementality.**  Central to our bidding optimization problem is the notion of *"conversion incrementality"* [18]. Intuitively, incrementality captures how much *increase* the advertiser's ads can boost the user's conversion rates. Notably, the reward of showing an ad is the incrementality

---

[3]In reality, $H$ is usually at the scale of around 50 (the number of times an advertiser would like to interact with a user), whereas the number of similar Internet users (i.e., episodes) $T$ is of the scale of millions.

while not the conversion rate itself. This is very different from the previous *learning to bid* literature since they assume the conversion (or value) comes independently with the previous ad placements (i.e., from oblivious adversary or an underlying distribution). This is why previous studies almost all use the (contextual) bandits setup, while not MDP.

We now formally describe the model of incrementality bidding, following [18]. Within any episode $t \in [T]$, each round $h \in [H]$ has an *incrementality rate function* $d_h(\tau; A_h, \theta_h)$ for any continuous time $\tau \in [h, \infty)$ which is the "density" of the conservation incrementality at any time $\tau \in [h, \infty)$. Intuitively, $d_h(\tau; A_h, \theta_h)$ models how much an ad shown at round $h$ boosts the conversion rate density at any future time $\tau > h$. Notably, the integral of this rate function $d_h(\tau; A_h, \theta_h)$ does not need to be 1 as there may be many conversions per user impression. We assume that $d_h(\tau; A_h, \theta_h)$ depends on an to-be-learnt unknown parameter $\theta_h$ and additionally a subset $A_h \subset [h]$ containing all previous rounds, at which our learner's ads were shown to the corresponding user of this episode. Let $\theta = (\theta_1, \theta_2, \cdots, \theta_H)$ denote all the to-be-learnt parameters.

It is generally intractable to learn an arbitrary rate function $d_h(\tau; A_h, \theta_h)$. Next, we introduce a natural *parameterized family* of functions for $d_h(\tau; A_h, \theta_h)$, with to-be-learnt parameters, in order to capture advertising applications. First, we have the following assumption for $d_h$,

**Assumption 1** (Markovian Incrementality). *$d_h$ is only affected by the learner's last ad impression before $h$, rather than the entire history $A_h$ of the learner's advertising performance. Formally, at current round $h$, suppose the last round at which the learner's ad was shown is $h - l$, then the incrementality function $d_h(\tau; A_h, \theta_h) = d_h(\tau; \ell, \theta_h)$.* [4]

The above assumption is built upon the sense that the users normally weighs more on the recently visited ads and are generally memory-less. Moreover, we assume each incrementality rate function $d_h(\tau; \ell, \theta_h)$ for any $h \in [H]$ has the following parametric form:

$$d_h(\tau; \ell, \theta_h) = \begin{cases} \beta_h(\ell)\lambda_h e^{-(\tau-h)\lambda_h} & \forall \tau \in [h, \infty), \\ 0 & otherwise, \end{cases} \tag{1}$$

in which $\theta_h = \{\beta_h(\ell)\}_{\ell=1}^{h-1} \cup \{\lambda_h\}$ contain all the parameters at round $h$ of the given IB instance. In other words, the incrementality rate is a re-scaled exponential distribution and the re-scaling is due to fact that each user impression may lead to more (or less) than one user conversions. In our RL formulation of the problem, these are all the to-be-learnt parameters. The Markovian property is reflected in $\beta_h(\ell)$. Note that $\int_{\tau=h}^{\infty} d_h(\tau; \ell, \theta_h)d\tau = \beta_h(\ell)$. So $\beta_h(\ell)$ can be viewed as the expected number of conversions triggered by the ad impression at round $h$, given the last time for showing the learner's ad is $\ell$ rounds before. In this paper, we assume $\forall h, \ell, c_\beta \leq \beta_h(\ell) \leq C_\beta$ for some positive constants $c_\beta$ and $C_\beta$. Without loss of generality, we assume $C_\beta \leq 1$. [5]

**Remark 1.** *The assumption of exponential decaying rate in Equation* (1) *is not necessary for our approach to work, and is primarily for the exposition. Our techniques can be applied to most parametric family of rate functions such as truncated exponential, logistic, Beta, and Gamma distributions [18], as long as the CDF function is invertible.*

**From conversion incrementality to realized conversions.** While incrementality captures how the rate of conversions increases, it did not model how conversions are realized in reality. This would be important when we introduce our reinforcement learning problem — after all, in real applications, the only observations our learner (an advertiser) can see are the realized conversions at different time, rather than the continuous conversion rate function. To capture this, we adopt the standard assumption that conversions are realized based on a continuous-time inhomogeneous *Poisson process* [18]. More specifically, let the *ordered* subset $W = \{w_1, w_2, \cdots, w_n\} \subseteq [H]$ denote the rounds at which the learning agent wins within the episode (for convenience, let $w_0 = 0$). Then the conversions arrive according to a Poisson process with a time-varying rate $r(\tau)$, defined as follows

---

[4] It is possible to have more features of the last ad impressions (instead of just the time gap $\ell$) in the state and our analysis will still go through as long as the number of states is still finite. For theoretical simplicity, in this paper, we focus on the case that the state is just $\ell$.

[5] In this paper, we assume $\beta_h(\ell)$ to be strictly positive, however, the conversion incrementality can be negative in practice, e.g., keep seeing the same ads very often may decrease the conversion from the users. Indeed, our results still hold when allowing negative $\beta_h(\ell)$, as long as $\beta_h$ and the time-varying rate $r(\tau)$ are bounded from zero. See the discussion in the proof of Theorem 1 in Appendix B.

$$r(\tau; w, \theta) = \sum_{i=1}^{n} d_{w_i}(\tau; w_i - w_{i-1}, \theta_{w_i}), \qquad \forall \tau \in [0, H]. \tag{2}$$

More details of inhomogeneous Poisson process is deferred to Appendix A.

**The optimization problem of bidding for incrementality.**  While the above incrementality modeling of user conversions originates from the influential idea of [18], our following formulation of the optimal bidding problem under their user modeling is new to the best of our knowledge, and so is its reinforcement learning solutions presented in this paper for addressing the situations with unknown environment parameters.

Following current practice, we assume online ad opportunities are sold to advertisers by auctions, and our learner is one of these advertisers. For the ease of presentation, we restrict our descriptions to the widely adopted *second price auctions*; however, all our results can be easily adapted to any single-item auction format without entry fees (e.g., the first price auction). Let $\mathcal{B}$ denote the bidding space, which may be continuous. Notably, $0 \in \mathcal{B}$, meaning the bidder/learner can choose to not participate this round's auction. Moreover, denote the distribution of the highest bid among *other* bidders (HOB) at round $h$ as $F_h$ (the cumulative distribution function or CDF). For any episode $t$ (corresponding to an Internet user), at each round $h$, the learner submits a bid $b \in \mathcal{B}$ and wins the opportunity to show its ad to the user if her bid is the highest. Let $p_h : \mathcal{B} \to \mathbb{R}_{\geq 0}$ represents the average (conditional) payment function of the learner if she wins. Let $v$ denote the learner's average value per conversion, which is known to the learner. We assume $v$ is bounded, w.l.o.g., within $[0, 1]$. Moreover, we can assume $\mathcal{B} \subseteq [0, 1]$. Then for any bid $b \in \mathcal{B}$, given the current state $\ell$, the immediate expected utility of the learner at round $h$ (for any episode) is,

$$u_h(b; \ell, \theta_h) = F_h(b) \cdot \left( \int_h^\infty v d_h(\tau; \ell, \theta_h) d\tau - p_h(b) \right) = F_h(b) \cdot (\beta_h(\ell) \cdot v - p_h(b)) \tag{3}$$

The astute reader may already find the expected utility function of the learner doesn't depend on parameter $\lambda_h$. This raises the question why we still need to know the function class of $d_h$ (i.e., exponential function) and why we still need to learn parameter $\lambda$. As we will see later in Section 4, due to the complexity of inhomogeneous Poisson process, it is crucial to know the function class of $d_h$ and we have to learn parameters $\lambda$ first in order to get a good estimator to $\beta$ afterwards. In second price auctions, $p_h(b)$ is the expected second highest bid conditioning on winning, which can be computed as follows

$$p_h(b) = \frac{1}{F_h(b)} \int_0^b v dF_h(v) = b - \frac{1}{F_h(b)} \int_0^b F_h(v) dv. \tag{4}$$

As mentioned above, our techniques apply to other auction formats as well. For example, in first price auctions, $p_h(b) = b$ has an even simpler format.

## 3  The Optimal Policy of IB and its Reinforcement Learning

While it may not appear obvious at the first glance, we explain how IB can be reduced to a finite-horizon Markov Decision Process (MDP) in this section and then introduce the corresponding reinforcement learning problem for IB when the environment parameters — namely, the incrementality parameters $\theta_h = \{\beta_h(l)\}_{l=1}^{h-1} \cup \{\lambda_h\}$ and HOB distribution $F_h$ — are unknown.

### 3.1  The Offline Optimal Policy of IB

We now formulate the optimal planning problem for the IB problem in its offline version, i.e., when all environment parameters are known. We start by defining what is a policy for the IB problem.

**Definition 1** (Policy). *A deterministic (dynamic) policy is a sequence of mappings $\pi = (\pi_1, \cdots, \pi_H)$, in which mapping $\pi_h : [h] \to \mathcal{B}$ maps any $\ell \in [h]$ to a bid. Here, $\ell$ is chosen such that the last winning of the learner happens at round $h - \ell$.*

Note that it suffice to consider policies that depend only on the number of consecutive loses $\ell$ due to the Markovian property of incrementality as mentioned above.

**The MDP re-formulation of `IB`.** The MDP will have $H$ *states*, with *state* $\ell \in [H]$ denoting that the last winning round is $\ell$ rounds ago from current round. The *action* set is $\mathcal{B}$, containing all possible bids. Both the state and action space are the same for each round $h$, however the transition and rewards are different for different rounds. Specifically, at round $h$, the transition probability from state $l$ and action $b$ is to enter state 1 at round $h + 1$ with probability $F_h(b)$, i.e., the probability of winning the auction, and to enter state $\ell + 1$ with probability $1 - F_h(b)$. The immediate reward from transitioning is $r_h(\ell, b, 1) = \int_h^\infty v(\tau) d_h(\tau; \ell, \theta_h) d\tau - p_h(b)$ for tuple $(\ell, b, 1)$ (the winning case) and $r_h(\ell, b, \ell + 1) = 0$ for tuple $(\ell, b, \ell + 1)$ (the losing case). This last statement depends crucially on the independence of the incrementality across different winning rounds. The initial state of this MDP is always the lowest state, i.e., $\ell_1 = 1$.

Consequently, the optimal policy in `IB` can be computed via standard dynamic programming. Specifically, when the action set $\mathcal{B}$ has finite support (i.e., the advertiser has finite bids to choose from), the following proposition follows from the fact that MDPs with finite states, actions and horizons can be solved efficiently by dynamic programming [1].

**Proposition 1.** *The offline optimal policy $\pi$ for any `IB` instance can be computed by dynamic programming in* $poly(|\mathcal{B}|, H)$ *time.*

We remark that even when $\mathcal{B} \subseteq \mathbb{R}$ is continuous with upper bound $B$, an $\epsilon$-optimal policy can be computed in $poly(\frac{B}{\epsilon}, H)$ time since the reward function is continuous in $b \in \mathcal{B}$ and an $\epsilon$-optimal action can be found at each round during the backward induction by discretizing the entire line of bid into $\epsilon$ segments. This $\epsilon$-optimal action choices lead to an approximately optimal policy [1].

## 3.2 Reinforcement Learning of `IB` with Unknown Parameters

We now turn to the much more interesting and realistic situation of the online learning, particularly reinforcement learning, of the optimal `IB` solution. While we have shown that the offline version of `IB` problem can be viewed as an MDP, its online counterpart cannot be similarly tackled by standard reinforcement learning techniques due to the following novel challenges specific to the `IB` problem:

- **Challenge 1: mixed reward feedback.** Any realized learner reward, arising from a user conversion, is naturally a *mixture* effect of all the previous winning rounds, as modeled by the accumulated incrementality rate in Equation (2). This is in contrast to standard MDP, in which the reward of this round is realized independently, conditioned on its state. It is not difficult to see that the loglikelihood of our mixed rewards model is non-convex, which renders the standard RL techniques based on maximum likelihood estimation for reward parameter estimation not applicable.
- **Challenge 2: delayed reward realizations.** Moreover, the conversions in the `IB` problem follow a intricate Poisson process with heterogeneous rate and thus has delays. That is, showing an ad may not immediately lead to conversions. Such lack of immediate reward feedback renders standard Q-learning style of algorithms not easily applicable to our setup since we cannot use immediate reward feedback to update Q-functions any more.

To overcome the first challenges above, we design a novel parameter estimation methods based on moment matching to estimate $\theta_h$'s from mixed reward feedback, and prove its convergence rate in Section 4. To address the second challenge above, we design a model-based no-regret RL algorithm that integrate the parameter estimation above, and analyze its regret in Section 5.

**Learner's Observations.** We now formalize our learning setup. We consider the episodic online learning setting, where the learner does not know the `IB` parameters $\theta_h = \{\beta_h(\ell)\}_{\ell=1}^{h-1} \cup \{\lambda_h\}$ for each $h \in [H]$ neither the distribution of the highest *other* bid (HOB) $F_h$. Following the typical practice, we assume that the learner can always observe the realized HOB at any round $h$ in each episode, denoted as $m_h^t$, regardless whether the learner wins or loses. This is usually termed as "*full information feedback*" in the literature of *learning to bid*.[6] Moreover, the learner also knows the set of its winning rounds denoted by an *ordered* set $\mathcal{W}^t \subset [H]$ as well as the set of the time stamps of all conversions $\mathcal{C}^t$ (a subset of $\mathbb{R}$) from each episode.

---

[6]This assumption is practical. First, in online ad systems, the manual bidder can always get this feedback (also known as "minimum bid to win") no matter she wins or not, e.g., in Google Ad exchange platform [7]. Second, for the auto bidding algorithms, they are designed by the platform and it does know the realized bids at each round.

In the online learning setting, the learner aims to design a bidding algorithm $\mathcal{A}$ to minimize the expected regret defined in the following,

$$\text{Regret}(T) = T \cdot \text{OPT}(\theta, F) - \mathbb{E}_{(\pi^1, \pi^2, \cdots, \pi^T) \sim \mathcal{A}} \left[ \sum_{t=1}^{T} R(\pi^t; \theta, F) \right], \tag{5}$$

where $\text{OPT}(\theta, F)$ represents the optimal expected utility achieved by the learner at each episode if the ground-truth parameters $\theta$ and $F$ are given (recall that the MDP always starts from state $\ell = 1$), $\mathbb{E}$ represents the expectation over the randomness of algorithm $\mathcal{A}$, and $\pi^t$ is the policy in the $t^{\text{th}}$ episode generated by the algorithm $\mathcal{A}$. Here, we slightly abuse the notation to denote $R(\pi; \widehat{\theta}, \widehat{F})$ as the expected utility achieved by policy $\pi$ for estimated parameters $\widehat{\theta}$ and $\widehat{F}$.

$$R(\pi; \widehat{\theta}, \widehat{F}) = \mathbb{E}_{(\ell_1, \cdots, \ell_H) \sim \widehat{P}^\pi} \left[ \sum_{h=1}^{H} \left( \widehat{\beta}_h(\ell_h) \cdot v - \widehat{p}_h(\pi_h(\ell_h)) \right) \cdot \widehat{F}_h(\pi_h(\ell_h)) \right], \tag{6}$$

where $\widehat{P}^\pi$ represents the joint distribution of states $(\ell_1, \cdots, \ell_H)$ induced by a bidding policy $\pi$ in the MDP formulated in Section 3.1 with estimated parameters $\widehat{\theta}$ and $\widehat{F}$. Note, for second price auctions, $\widehat{p}_h(b) = b - \frac{1}{\widehat{F}_h(b)} \int_0^b \widehat{F}_h(v) dv$ (replacing $F$ by $\widehat{F}$ in Eq. (4)). In other words, $R(\pi; \widehat{\theta}, \widehat{F})$ captures the expected total reward in the MDP when the reward function and transition probability are parameterized by $\widehat{\theta}$ and $\widehat{F}$.

**Remark on a non-degeneracy assumption on the learner.** We assume that the learner's bidding space $\mathcal{B}$ is upper bounded away from extreme bids. More formally, there exists a small constant $c_0$ such that $F(b) \leq 1 - c_0$ for any $b \in \mathcal{B}$ (i.e., no feasible bids guarantee sure winning). We call such learner $c_0$-*bounded*. There are two reasons for making this assumption. The first is a technical reason: it makes sure that the learner always has at least $c_0$ probability to lose, which turns out to be crucial to estimate the incrementality parameters, as we do in the next section. This is because incrementality captures the *difference* between winning and losing. If the probability of losing is extremely small, there is generally no way to accurately estimate their differences (though it turns out that we can afford small winning probabilities). The second reason for assuming $c_0$-bounded learners is realistic motivations since in auto bidding practice, bidders typically explicitly specify upper bounds of their bids, under which the bidder's winning probability is almost always capped strictly below 1.

## 4 Incrementality Parameter Estimation via Pairwise Moment-Matching

Though maximum likelihood estimation (MLE) is a natural and straightforward parameter estimation strategy, it comes with many drawbacks we want to avoid. First, the MLE criterion under our model of mixed rewards is not a convex problem and optimizing the likelihood would suffer from multiple local optimal or saddle point issues. The theoretical claims may not match the true algorithm performance. Secondly, the computational efficiency of the MLE problem is worse than what we will introduce next, which makes it less attractive in large-scale problems. Our algorithm is carefully tailored for estimating parameters of our model, given the observed trajectories of first $t$ rounds. Specifically, the trajectory of each episode $t \in [T]$ can be described by two sequences: (1) the time stamps $\mathcal{C}^t$ of realized conversions within time $[0, \infty]$; and (2) the *ordered* subset $\mathcal{W}^t = \{w_1^t, \cdots, w_{n_t}^t\} \subseteq [H]$ which contains all the winning steps within this episode (for convenience, we extend the above notations and by letting $w_0^t = 0, w_{n_t+1}^t = H + 1$), where $n_t$ denotes the number of winning events (showing the ad) in the episode $t$. Equivalently, we can also represent $\mathcal{W}^t$ by a $H$-length binary sequence $\zeta^t \in \{0, 1\}^H$ such that $\zeta_h^t = 1$ if and only if episode $t$ wins at time $h$. Notably, both $\mathcal{C}^t$ and $\mathcal{W}^t$ may be empty if there is no conversion nor winning. Our estimation is based on an online algorithm: from a high-level perspective, it progresses by time intervals $[h, h+1)$ sequentially to produce estimates of parameters $\theta_h$, by pairwisely coupling episodes and matching statistical moments. Therefore, we call it the *pairwise moment-matching* (PAMM) algorithm, described in Algorithm 1.

The PAMM algorithm is specifically designed for estimating parameters from mixed reward signals, each drawn from a Poisson process. Its primary advantage lies in its online nature, guaranteed convergence and efficient computation. This is achieved by the following observations. First, all information about reward scale parameter $\beta_h$ and reward variance parameter $\lambda_h$ is intrinsically

---

**Algorithm 1** Pairwise Moment-Matching (`PAMM`) Algorithm

---

1: **Input**: A sample of episodes $(\mathcal{C}^t, \mathcal{W}^t), t \in [T]$ up to time $[0, h+1)$.

2: **Output:** the estimates of parameter $\theta_h$, denoted by $\{\widehat{\beta}_h(\ell), \forall \ell \in [h]; \widehat{\lambda}_h\}$.

3: For any $t$, define $N_h^t$ and $\overline{N}_h^t$ to be the number of conversions of episode $t$ within $[h, h+\frac{1}{2})$ and $[h+\frac{1}{2}, h+1)$, respectively.

4: **for** For each given $h$ of interest, **do**

5:     Define $n_h$ to be the total number of episodes with winning bid at time $h$. Define $\Psi = \{0,1\}^{h-1}$ be the set of all possible winning sequences before time $h$. For each winning sequence $\phi \in \Psi_h$, define

$$\Omega_\phi = \{t \in T : \zeta_h^t = 1, \zeta_j^t = \phi_j, 1 \le j \le h-1\}, \Lambda_\phi = \{t \in T : \zeta_h^t = 0, \zeta_j^t = \phi_j, 1 \le j \le h-1\}.$$

    Moreover, define $n_\phi = |\Omega_\phi|$, $n'_\phi = |\Lambda_\phi|$, and $\widetilde{n}_\phi = \left(\frac{n_\phi^{-1} + (n'_\phi)^{-1}}{2}\right)^{-1}$ as the harmonic mean of $n_\phi$ and $n'_\phi$.

6:     For each $\phi \in \{0,1\}^{h-1}$,

$$X_\phi = \frac{1}{n_\phi} \sum_{t \in \Omega_\phi} N_h^t - \frac{1}{n'_\phi} \sum_{t \in \Lambda_\phi} N_h^t \quad \text{and} \quad Y_\phi = \frac{1}{n_\phi} \sum_{t \in \Omega_\phi} \overline{N}_h^t - \frac{1}{n'_\phi} \sum_{t \in \Lambda_\phi} \overline{N}_h^t. \quad (7)$$

7:     For each $\phi \in \{0,1\}^{h-1}$, define

$$\alpha_\phi = \frac{\widetilde{n}_\phi}{\sum_{\phi' \in \Psi_h} \widetilde{n}_{\phi'}} \quad (8)$$

    and set

$$\widehat{\mu}_h = \sum_{\phi \in \Psi_h} \alpha_\phi X_\phi \text{ and } \widehat{\eta}_h = \sum_{\phi \in \Psi_h} \alpha_\phi Y_\phi. \quad (9)$$

    Estimate $\lambda_h$ by

$$\widehat{\lambda}_h = 2(\log \widehat{\mu}_h - \log \widehat{\eta}_h). \quad (10)$$

8:     **for** $1 \le \ell \le h$ **do**

9:         Define the $\ell$th slice of $\Psi_h$ as $\Psi_{h,\ell} = \{\phi \in \Psi_h : \phi_{h-\ell} = 1, \phi_k = 0, h - \ell < k \le h - 1\}$. For each winning sequence $\phi \in \Psi_{h,\ell}$, still define $X_\phi$ and $Y_\phi$ following (7).

10:        Define

$$\widehat{\mu}_{h\ell} = \sum_{\phi \in \Psi_{h,\ell}} \alpha_{\phi\ell} X_\phi \text{ and } \widehat{\eta}_{h\ell} = \sum_{\phi \in \Psi_{h,\ell}} \alpha_{\phi\ell} Y_\phi. \quad (11)$$

       where $\alpha_{\phi\ell}$ is defined following (8) after replacing $\Psi_h$ by $\Psi_{h,\ell}$.

11:        Estimate $\beta_h(\ell)$ by

$$\widehat{\beta}_h(\ell) = \frac{\widehat{\mu}_{h\ell}^2}{\widehat{\mu}_{h\ell} - \widehat{\eta}_{h\ell}}. \quad (12)$$

12:     **end for**

13: **end for**

---

reflected only in the "differential behaviors" for those episodes which had conversions vs. those episodes which did not have conversions at time step $h$, conditioning their matched history before time $h$ as defined in the sample matching step. This is due to the Markovian property for the Poisson process, and our moment matching is hinged into this property. Second, for the Poisson distribution (as a member of the exponential family), the method of matching the canonical parameter (mean) gives an equivalent estimator as the MLE in a single Poisson case (but not necessary in our more complicated model). These two ideas motivate our procedure, by matching the episodes, we precisely locate the needed Poisson signals and solve the estimators by the mean value structure. Because of this, the `PAMM` can recover the parameters with essentially optimal rate (in the online sense), even though we do not use the full likelihood that would result in non-convexity and an unpleasant gap between the theory and the practical algorithm.

In practice, terms in (10) and (12) may result in $\pm\infty$, but such an event only happens with probability tending to zero (which is why Theorem 1 has high probability guarantees). Standard numerical protection can be introduced in practice which will not change our theoretical claims. Moreover, from the implementation perspective, one start from the first interval $[0,1)$ and move forward by running PAMM for each $[h, h+1)$. In each new interval, the matching step can be directly carried over by further refining the matching of the previous interval. As a result, the whole estimation procedure only involves one sweep of the data without the commuting back and forth between different time intervals. We have introduced the PAMM algorithm in the current form as a self-contained algorithm for estimating the inhomogeneous Poisson process, which can be of independent interest. However, the steps of PAMM can be easily embedded in our online learning algorithm presented in Section 5 without losing its validity.

**Theorem 1** (Estimation Errors with Finite Samples). *Suppose there exist positive constants $C_\lambda$ and $c_\beta$, such that $c_\lambda \leq \lambda_h \leq 1/c_\lambda$ and $\beta_h(\ell) \geq c_\beta$ for any $h$ and $\ell$, and the learner is $c_0$-bounded. Also assume that the expected number of total conversions for each episode within each interval $[h, h+1)$ is always bounded above by a constant $C_T$. Let $n_h$ be the total number of winning episodes at time $h$ and $n_h(\ell)$ be the total number of winning episodes at time $h$ with state $\ell$. Let $\widehat{\lambda}_h$ and $\widehat{\beta}_h$ be the estimators obtained from Algorithm 1. Then, for any $0 \leq h < H$ and any $0 \leq \ell < h$, we have the following guarantees for PAMM:*

1. *For any $4e^{-C_0 n_h} < \delta < 1/2$, with probability at least $1 - \delta$, $|\widehat{\lambda}_h - \lambda_h| \leq C_1 \sqrt{\frac{\log 1/\delta}{n_h}}$.*

2. *For any $4e^{-C_0 n_h(\ell)} < \delta < 1/2$, with probability at least $1 - \delta$, $|\widehat{\beta}_h(\ell) - \beta_h(\ell)| \leq C_2 \sqrt{\frac{\log 1/\delta}{n_h(\ell)}}$.*

*In the above results, $C_0 = \frac{c' c_0}{2}, C_1 = 4\sqrt{\frac{C_T^2}{c_\beta^2 (1 - e^{-c_\lambda/2})^2 c' c_0}}$ and $C_2 = 8\sqrt{\frac{2C_T^2}{c' c_0}}$, where $c'$ is the Bernstein constant (see Lemma 1).*

## 5 The Full RL Algorithm and its Regret Analysis

We propose a reinforcement learning algorithm by incorporating our parameter estimation method PAMM and the optimal offline planning for the MDP mentioned in Section 3 to learn parameters $\theta, F$ as well as the conversion incrementality and decide the bid at each round.

Our algorithm adopts a UCB-style procedure to handle the exploration-exploitation tradeoff in incrementality bidding, which is summarized in Algorithm 2. The algorithm adopts pure explorations in several beginning episodes so that we can get enough ad impressions for each state in every round. These pure explorations are achieved by setting bids appropriately such that we can enforce to explore one $(h, \ell)$ pair in each exploration (Line 4 in Algorithm 2)[7]. In particular, we need $O(H^2 \log(T/\delta))$ explorations to make sure $n_h(\ell) \geq \frac{\log(4T/\delta)}{C_0}$ so that we can provide a regret bound with high probability, where $C_0$ is a positive constant explicitly defined in Theorem 1. Recall $n_h(\ell)$ is the total number of winnings at round $h$ given the state is $\ell$.

After pure explorations, at each episode $t$, we use the PAMM algorithm to get an estimator $\widehat{\theta}^t$ (resp. $\widehat{\beta}^t$) given the observed ad impression time $\mathcal{W}^t$ and the conversion time $\mathcal{C}^t$ up to episode $t$. In addition, we can estimate $F$ easily using empirical distribution function and this implies we can get a uniform convergence for the transition probabilities for any action $b \in \mathcal{B}$ given any state $\ell$. Then we apply UCB-style RL algorithm with estimated parameters $\widehat{\theta}^t$ to decide the policy in next episode. Specifically, given the estimated $\widehat{\beta}^t, \forall h \in [H]$, we construct a confidence region centered at $\widehat{\beta}^t$ and we do offline planning for the *optimism* in the confidence region to get the policy in next episode.

**Remark 2** (Computational Efficiency of Alg. 2). *An important observation is that $R(\pi; \widehat{\theta}, \widehat{F})$ is non-decreasing in $\widehat{\beta}_h(\ell)$ (see Eq. (6)) for any fixed $\pi$ and $\widehat{F}$. Therefore, the optimal choice of $\widehat{\beta}$ in*

---

[7]When $\ell = h$, setting $b_0^t = 1$ means a "fake" winning at round 0 and it is consistent with our assumption that initial state $\ell_1 = 1$.

---

**Algorithm 2** Online Bidding algorithm

---

1: **Input**: Given $\delta$, $H$, $T$, and $C_0, C_2$ defined in Theorem 1. Set $n_h(\ell) = 0, \forall h \in [H], \ell \in [h]$.
2: **for** Episode $t = 1, \cdots, T$ **do**
3:     **if** $\exists h, \ell, n_h(\ell) < \frac{\log(4T/\delta)}{C_0}$ **then**
4:         Set bids, s.t. $b_h^t = b_{h-\ell}^t = 1$, and $b_{h'}^t = 0, \forall h' \neq h, h - \ell$.     [Pure Exploration]
5:     **else**
6:         For $h \in [H]$, observe state $\ell_h^t$ and set the bid $b_h^t = \pi_h^t(\ell_h^t)$.
7:     **end if**
8:     Observe the set of the ad impression time $\mathcal{W}^t$, the set of the conversion time $\mathcal{C}^t$, and the vector of HOBs $m^t = \{m_h^t, \forall h \in [H]\}$.
9:     Update $n_h(\ell)$ given $\mathcal{W}^t$.
10:     Update $\widehat{\theta}^t = \{\widehat{\beta}_h^t(\ell), \forall \ell \in [h]; \widehat{\lambda}_h^t\}$ through PAMM method (Algorithm 1) given $(\mathcal{W}^t, \mathcal{C}^t)$.
11:     Update $\widehat{F}_h^t(b) = \frac{1}{t}\sum_{s=1}^t \mathbb{I}\{m_h^s \leq b\}, \forall b$.
12:     **if** $\forall h \in [H], \ell \in [h], n_h(\ell) \geq 1$ **then**
13:         Construct confidence region $\widehat{\text{CR}}^t$ for parameters $\theta$, s.t.

$$\widehat{\text{CR}}^t = \left\{ (\widehat{\beta}, \widehat{\lambda}) : \forall h, |\widehat{\beta}_h(\ell) - \widehat{\beta}_h^t(\ell)| \leq C_2 \sqrt{\frac{\log(T/\delta)}{n_h(\ell)}} \right\}$$

14:         Compute the bidding policy $\pi^{t+1}$ for $(t+1)$th episode, s.t.,

$$\pi^{t+1} = \underset{\pi}{\text{argmax}} \max_{\widehat{\theta} \in \widehat{\text{CR}}^t} R(\pi; \widehat{\theta}, \widehat{F}^t) \tag{13}$$

15:     **end if**
16: **end for**

---

*Eq. (13) in Algorithm 2 is to set $\widehat{\beta}_h(\ell) = \widehat{\beta}_h^t(\ell) + C_2\sqrt{\frac{\log(1/\delta)}{n_h^t(\ell)}}$, and $\pi^{t+1}$ will be the optimal policy for the MDP parameterized by $\widehat{F}^t$ and $\widehat{\beta}$, and thus can be computed efficiently (Proposition 1).*

**Regret Analysis.** Combining the convergence rate analysis from Theorem 1 and a novel regret decomposition technique (necessary for our particular setup), we are able to prove the regret guarantee of our online bidding algorithm. Our regret bound is summarized in Theorem 2, the proof of which is deferred to Appendix C.

**Theorem 2.** *Under Assumption 1, for any fixed $\delta$, we have with probability at least $1 - \delta$, the regret achieved by our Algorithm 2 is bounded by*

$$\text{Regret}(T) \leq O(H^2\sqrt{\log(HT/\delta)T} + H^2 \log(T/\delta)).$$

# 6 Discussions and Future Work

Our RL algorithm lies in the model-based RL algorithm literature. The best known regret bound for model-based RL is $\widetilde{O}(\sqrt{H^2 SAT})$ [2], where $S$ is the number of states and $A$ is the number of actions. Since $S = H$ in our case, our regret bound is slightly worse from the best possible by an factor of $\sqrt{H}$, but is independent of the number of actions. This may be due to the complexity of our model. Whether we can close this gap is an interesting open question. As we mentioned earlier, our algorithm can be extended to handle continuous bidding space and our regret analysis still works. In this paper, we assume the conversion rate function $d_h$ doesn't depend on the shown ads of other bidders but only on the learner's *last* ad impression. Relaxing these assumptions may result in very interesting future work.

## Acknowledgements

T. Li is supported in part by the NSF grant DMS-2015298 and 3Caverliers award from the University of Virginia. H. Xu is supported in part by an ARO award W911NF-23-1-0030.

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
