# Incrementality Bidding via Reinforcement Learning under Mixed and Delayed Rewards

# Appendix

## A  Formal Definition of Inhomogeneous Poisson Process

The inhomogeneous Poisson (point) process is a Poisson point process with a Poisson parameter set as some time-dependent function $r(\tau)$. In particular, the expected number of points observed in a time interval $[a, b]$ is $\Lambda(a, b) = \int_a^b r(\tau)d\tau$. Let $N(a, b)$ represent the number of points of inhomogeneous Poisson process with intensity function $r(t)$ occurring in the interval $[a, b]$, then the probability of $n$ points existing in the interval $[a, b]$ is given by,

$$P(N(a, b) = n)\frac{\Lambda(a, b)^n}{n!}e^{-\Lambda(a,b)}$$

In this paper, the points mean the conversions and the time-dependent intensity function $r(\cdot)$ is defined in Eq. (2) and it depends on the realization of the conversions and parameter $\theta$.

## B  Poisson Process Estimation

**Lemma 1** (Bernstein's Inequality [24])**.** *Suppose $X_1, \cdots X_n$ are independent, mean-zero, sub-exponential random variables, and $a = (a_1, \cdots, a_n)$ is an $n$ dimensional constanst vector. Then, for every $\epsilon > 0$, we have*

$$\mathbf{Pr}(|\sum_{i=1}^n a_i X_i| \geq \epsilon) \leq 2\exp\left(-c'\min\left(\frac{\epsilon^2}{\max\|X_i\|_\psi^2\|a\|^2}, \frac{\epsilon}{\max_i\|X_i\|_\psi\|a\|_\infty}\right)n\right)$$

*where $\|X_i\|_\psi$ is the sub-exponential norm (Orlicz norm) of $X_i$ and $c'$ is an absolute constant (Bernstein constant). $\|a\|$ is the Euclidean norm and $\|a\|_\infty = \max_i |a_i|$.*

*Proof of Theorem 1.* Denote $\Psi_h = \{0, 1\}^{h-1}$. We first introduce the main idea of the the PAMM algorithm. Since conversions follow the Poisson process, we have $N_h^t \sim$ Poisson$(\int_h^{h+1/2} r(\tau; w^t, \theta_h)d\tau)$ and $\overline{N}_h^t \sim$ Poisson$(\int_{h+1/2}^{h+1} r(\tau; w^t, \theta_h)d\tau)$ independently.

Note that for all $t \in \Omega_\phi$ and $\Lambda_\phi$, their history winning records before $h$ are exactly the same. Suppose $s^\phi = \max\{j : j \leq h-1, \phi_j = 1\}$. It is not difficult to see that

$$\mathbb{E}(X_\phi) = \int_h^{h+1/2} r(\tau; w^t, \theta)d\tau - \int_h^{h+1/2} r(\tau; w^{t'}, \theta)d\tau$$

$$= \int_h^{h+1/2}\sum_{w_i^t \leq h-1} d_{w_i^t}(\tau; w_i^t - w_{i-1}^t, \theta_{w_i})d\tau + \int_h^{h+1/2} d_h(\tau; h - s^t, \theta_h)d\tau$$

$$- \int_h^{h+1/2}\sum_{w_i^{t'} \leq h-1} d_{w_i^{\phi(t)}}(\tau; w_i^{t'} - w_{i-1}^{t'}, \theta_{w_i^{t'}})d\tau$$

$$= \int_h^{h+1/2} d_h(\tau; h - s^t, \theta_h)d\tau = \beta_h(h - s^\phi)\left(1 - e^{-\lambda_h/2}\right) \tag{14}$$

where $t$ and $t'$ are two arbitrary episodes in $\Omega_h$ and $\Lambda_h$ (nonempty). Similarly, we have

$$\mathbb{E}(Y_\phi) = \beta_h(h - s^\phi)\left(1 - e^{-\lambda_h/2}\right)e^{-\lambda_h/2}. \tag{15}$$

Therefore, $\mathbb{E}(X_\phi)$ and $\mathbb{E}(Y_\phi)$ precisely locate the parameter of interest, $\theta_h$. By aggregating the information over different $\phi$, we can thus estimate $\{\beta_h(\ell), 0 \leq \ell \leq h\}$ and $\lambda_h$ based on the method of moments. Notice that $\lambda_h$ is shared for all episodes with the same $\phi$ while $\beta_h(\ell)$ is shared only

within a subset of $\Psi_h$, for a given $\ell$. Therefore, we will tailor the moment match operations according to different aggregation levels, as indicated in (10) and (12). Next, we proceed to study the theoretical property of the algorithm. Define

$$\mu_h = \sum_{\phi \in \Psi_h} \alpha_\phi \beta_h (h - s^\phi)(1 - e^{-\lambda_h/2})$$

and

$$\eta_h = \sum_{t \in \Psi_h} \alpha_\phi \beta_h (h - s^\phi) e^{-\lambda_h/2}(1 - e^{-\lambda_h/2}).$$

Taking the logarithm of the ratio between the two quantities, we get

$$\log(\mu_h/\eta_h) = \lambda_h/2.$$

Our estimator $\widehat{\lambda}_h$ is motivated by the above identity with the observation that

$$\mathbb{E}\,\widehat{\mu}_h = \mu_h; \quad \mathbb{E}\,\widehat{\eta}_h = \eta_h.$$

We start with deriving the concentration of $\widehat{\mu}_h$. Note that the terms $N_h^t$ in (7) are Poisson random variables, with their expectation upper bounded by

$$\int_h^{h+1/2} r(\tau; w^t, \theta_h) d\tau \le C_T.$$

Therefore, each of these items is a sub-exponential random variable with $\|N_h^t\|_\psi \le C_T$. To use Lemma 1, we can see that the $a$ vector corresponding to (9) reads as

$$\|a\|^2 = \sum_{\phi \in \Psi_h} \alpha_\phi^2 \left(\frac{1}{n_\phi} + \frac{1}{n_\phi'}\right)$$

$$= \sum_{\phi \in \Psi_h} \frac{(\frac{1}{\frac{1}{n_\phi} + \frac{1}{n_\phi'}})^2 (\frac{1}{n_\phi} + \frac{1}{n_\phi'})}{(\sum_{\phi \in \Psi_h} \frac{1}{\frac{1}{n_\phi} + \frac{1}{n_\phi'}})^2} = \frac{1}{\sum_{\phi \in \Psi_h} \frac{1}{\frac{1}{n_\phi} + \frac{1}{n_\phi'}}}$$

$$= \frac{2}{\sum_{\phi \in \Psi_h} \widetilde{n}_\phi}$$

where $\widetilde{n}_\phi$ is the harmonic mean of $n_\phi$ and $n_\phi'$. Also, we have

$$\|a\|_\infty = \frac{1}{\sum_{\phi \in \Psi_h} \frac{1}{\frac{1}{n_\phi} + \frac{1}{n_\phi'}}} \max_\phi \frac{1}{\frac{1}{n_\phi} + \frac{1}{n_\phi'}} \max\left(\frac{1}{n_\phi}, \frac{1}{n_\phi'}\right) \le \frac{2}{\sum_\phi \widetilde{n}_\phi}.$$

Define $\widetilde{n}_h = \frac{1}{\|a\|^2} = \frac{\sum_{\phi \in \Psi_h} \widetilde{n}_\phi}{2}$. With the $c_0$-bounded assumption and the exploration stage requirement for sample, it is easy to see that we have $\widetilde{n}_h \ge c_0 n_h$ (happening with probability tending to 1 regarding the randomness of bidding). Therefore, for sufficiently large $n_h$, by Lemma 1, we have,

$$\mathbf{Pr}(|\widehat{\mu}_h - \mu_h| > \epsilon) \le 2 \exp\left(-c' \min\left(\frac{\epsilon^2}{2C_T^2}, \frac{\epsilon}{2C_T}\right)\widetilde{n}_h\right) \le 2 \exp\left(-\widetilde{c} \min\left(\frac{\epsilon^2}{2C_T^2}, \frac{\epsilon}{2C_T}\right)n_h\right)$$

for constant $\widetilde{c} = c'c_0$. Assuming $\delta > \exp(-\frac{\widetilde{c}}{2}n_h)$, and setting $\epsilon = \sqrt{2}\frac{C_T}{\sqrt{\widetilde{c}}}\sqrt{\frac{\log 1/\delta}{n_h}}$, we will be able to use the sub-gaussian bound of the concentration, which gives

$$\mathbf{Pr}\left(|\widehat{\mu}_h - \mu_h| > \sqrt{\frac{2C_T^2}{\widetilde{c}}}\sqrt{\frac{\log 1/\delta}{n_h}}\right) \le 2\delta.$$

For notational convenience, we will denote this relation by

$$\widehat{\mu}_h = \mu_h + O_P\left(\sqrt{\frac{2C_T^2}{\widetilde{c}}}\sqrt{\frac{\log 1/\delta}{n_h}}\right) \tag{16}$$

where $O_P\left(\sqrt{\frac{\log 1/\delta}{n_h}}\right)$ refers to any generic term in the order of $\sqrt{\frac{\log 1/\delta}{n_h}}$ with probability at least $1 - 2\delta$.

Furthermore, by Taylor expansion, we have

$$\log(\widehat{\mu}_h) = \log \mu_h + \frac{\widehat{\mu}_h - \mu_h}{\mu_h} + o(|\widehat{\mu}_h - \mu_h|).$$

Since $\lambda_h \leq C_\lambda$ and $|\beta_h(\ell)| \geq c_\beta$, we have

$$|\mu_h| \geq c_\beta(1 - e^{-c_\lambda/2}) \tag{17}$$

in which the right hand side is a constant. Therefore, we have

$$|\log \widehat{\mu}_h - \log \mu_h| = O_P\left(2\sqrt{\frac{C_T^2}{c_\beta^2(1 - e^{-c_\lambda/2})^2\widetilde{c}}}\sqrt{\frac{\log 1/\delta}{n_h}}\right) = O_P\left(\sqrt{\frac{\log 1/\delta}{n_h}}\right).$$

Similarly, because

$$\eta_h \geq c_\beta e^{-\frac{1}{2c_\lambda}}(1 - e^{-c_\lambda/2}), \tag{18}$$

we can also show the same property for $\log \widehat{\eta}_h$ with

$$|\log \widehat{\eta}_h - \log \eta_h| = O_P\left(2\sqrt{\frac{C_T^2}{c_\beta^2(1 - e^{-c_\lambda/2})^2\widetilde{c}}}\sqrt{\frac{\log 1/\delta}{n_h}}\right).$$

Combining the results for $\log \widehat{\mu}_h$ and $\log \widehat{\eta}_h$, we know that

$$|\widehat{\lambda}_h - \lambda_h| = O_P\left(4\sqrt{\frac{C_T^2}{c_\beta^2(1 - e^{-c_\lambda/2})^2\widetilde{c}}}\sqrt{\frac{\log 1/\delta}{n_h}}\right).$$

Note that our proof allows negative values be $\beta_h(\ell)$. This is because the only places where we use the lower bound $c_\beta$ so far are (17) and (18). In general, as long as we can have a lower bound for $|\mu_h|$ and $|\eta_h|$, the proof still works.

For the estimation of $\beta_h(\ell)$, a similar approach can be taken, but the stratum of focus will be on the slide $\Psi_{h,\ell}$. First, notice that (14) and (15) remain valid for $\Psi_{h,\ell}$. Indeed, all of the $X_\phi$'s share the same expectation. Define

$$\mu_{h\ell} = \beta_h(\ell)(1 - e^{-\lambda_h/2}), \quad \eta_{h\ell} = \beta_h(\ell)(1 - e^{-\lambda_h/2})e^{-\lambda_h/2}.$$

Now the moment matching equation becomes

$$\beta_h(\ell) = \frac{\mu_{h\ell}^2}{\mu_{h\ell} - \eta_{h\ell}}.$$

In this case, we still have

$$\mathbb{E}\,\widehat{\mu}_{h\ell} = \sum_{\phi \in \Psi_{h,\ell}} \alpha_{\phi\ell}\beta_h(\ell)(1 - e^{-\lambda_h/2}) = \beta_h(\ell)(1 - e^{-\lambda_h/2})$$

and

$$\mathbb{E}\,\widehat{\eta}_{h\ell} = \sum_{\phi \in \Psi_{h,\ell}} \alpha_{\phi\ell}\beta_h(\ell)(1 - e^{-\lambda_h/2})e^{-\lambda_h/2} = \beta_h(\ell)(1 - e^{-\lambda_h/2})e^{-\lambda_h/2}.$$

Therefore, we just need to check the concentration bounds of $\widehat{\mu}_{h\ell}$ and $\widehat{\eta}_{h\ell}$. Define $\widetilde{h}(\ell) = \frac{\sum_{\phi \in \Psi_{h,\ell}} \widetilde{n}_\phi}{2}$ where $\widetilde{n}_\phi$ is the harmonic mean of $n_\phi$ and $n'_\phi$. With the assumption that $\widetilde{n}_h(\ell) \geq c_0 n_h(\ell)$, by the same derivation as for $\widehat{\mu}_h$, we can get

$$|\widehat{\mu}_{h\ell} - \mu_{h\ell}| = O_P\left(\sqrt{\frac{2C_T^2}{\widetilde{c}}}\sqrt{\frac{\log 1/\delta}{n_h(\ell)}}\right)$$

as long as $\delta > \exp(-\frac{\widetilde{c}}{2}n_h(\ell))$. Similarly,

$$|\widehat{\eta}_{h\ell} - \eta_{h\ell}| = O_P\big(\sqrt{\frac{2C_T^2}{\widetilde{c}}}\sqrt{\frac{\log 1/\delta}{n_h(\ell)}}\big).$$

Now, define $f(x, y) = \frac{x^2}{x-y}$. By multivariate Taylor expansion, we have

$$|f(\widehat{\mu}_{h\ell}, \widehat{\eta}_{h\ell}) - f(\mu_{h\ell}, \eta_{h\ell})| \leq \|\nabla f(\mu_{h\ell}, \eta_{h\ell})\|\sqrt{|\widehat{\eta}_{h\ell} - \eta_{h\ell}|^2 + |\widehat{\eta}_{h\ell} - \eta_{h\ell}|^2} + o(\sqrt{|\widehat{\eta}_{h\ell} - \eta_{h\ell}|^2 + |\widehat{\eta}_{h\ell} - \eta_{h\ell}|^2})$$

where $\nabla f(\mu_{h\ell}, \eta_{h\ell})$ is the gradient with

$$\|\nabla f(x, y)\| = \frac{|x|\sqrt{2x^2 - 4xy + 4y^2}}{(x-y)^2} \leq \frac{|x|(|x| + |x - 2y|)}{(x-y)^2}.$$

Substituting $\mu_{h\ell}$ and $\eta_{h\ell}$ into the gradient leads to

$$\|\nabla f(\mu_{h\ell}, \eta_{h\ell})\| \leq 2 - 2e^{-\lambda_h/2} \leq 2.$$

Therefore, we have

$$|\widehat{\beta}_h(\ell) - \beta_h(\ell)| = |f(\widehat{\mu}_{h\ell}, \widehat{\eta}_{h\ell}) - f(\mu_{h\ell}, \eta_{h\ell})| = O_P\big(8\sqrt{\frac{2C_T^2}{\widetilde{c}}}\sqrt{\frac{\log 1/\delta}{n_h(\ell)}}\big).$$

Notice that we have $C_1 = 4\sqrt{\frac{C_T^2}{c_\beta^2(1-e^{-c\lambda/2})^2 c' c_0}}$ and $C_2 = 8\sqrt{\frac{2C_T^2}{c' c_0}}$.

$\square$

## C   Proof of Theorem 2

To prove Theorem 2, we need several auxiliary lemmas. First, we provide a lemma to bound $R(\pi; \widehat{\theta}, \widehat{F}) - R(\pi; \theta, \widehat{F})$. The importance of this lemma is that we provide a reformulation of function $R(\pi; \widehat{\theta}, \widehat{F})$ by utilizing the fact that the conversion incrementality can only happen if the learner wins the opportunity to show the ad to the user. This observation makes our regret bound independent of the number of actions.

**Lemma 2.** *Given estimated parameters $\widehat{\theta}$ and $\widehat{F}$, for any bidding policy $\pi$, we have*

$$R(\pi; \widehat{\theta}, \widehat{F}) - R(\pi; \theta, \widehat{F}) \leq \mathop{\mathbb{E}}_{\mathcal{L} \sim \widehat{P}^\pi, o_h(\cdot) \sim Ber(\widehat{F}_h(\cdot))}\left[\sum_{h=1}^H |\widehat{\beta}_h(\ell_h) - \beta_h(\ell_h)| \cdot \mathbb{I}\big\{o_h\big(\pi(\ell_h)\big) = 1\big\}\right],$$

*where $\widehat{P}^\pi(\mathcal{L})$ is the probability of sequence $\mathcal{L} = (\ell_1, \ell_2, \cdots, \ell_H)$ induced by the MDP parameterized by $\widehat{\theta}$ and $\widehat{F}$ by adopting policy $\pi$ and $o_h(b)$ models whether the learner wins ($o_h(b) = 1$) or not ($o_h(b) = 0$) at round $h$ with bid $b$ (given HOB follows distribution $\widehat{F}_h$ at each round $h$).*

*Proof.* Recall, by definition of $R(\pi; \widehat{\theta}, \widehat{F})$ in Eq. (6), for any $\widehat{\theta}$ and $\widehat{F}$,

$$R(\pi; \widehat{\theta}, \widehat{F}) = \mathop{\mathbb{E}}_{\mathcal{L} \sim \widehat{P}^\pi}\left[\sum_{h=1}^H \big[\widehat{\beta}_h(\ell_h) \cdot v - \widehat{p}_h(\pi_h(\ell_h))\big] \cdot \widehat{F}_h(\pi_h(\ell_h))\right], \tag{19}$$

Let $o_h(b)$ denote the realized outcome of the learner at round $h$, s.t.,

$$o_h(b) = \begin{cases} 1 & \text{w.p.} & \widehat{F}_h(b) \\ 0 & \text{w.p.} & 1 - \widehat{F}_h(b) \end{cases} \tag{20}$$

Given the above reformulation of $R(\pi; \widehat{\theta}, \widehat{F})$, we can derive the difference between $R(\pi; \widehat{\theta}, \widehat{F})$ and $R(\pi; \theta, \widehat{F})$ for any bidding policy $\pi$ and any distribution of HOB $\widehat{F}_h$ at each round $h$,

$$
\begin{aligned}
R(\pi; \widehat{\theta}, \widehat{F}) - R(\pi; \theta, \widehat{F}) &= \mathop{\mathbb{E}}_{\mathcal{L} \sim \widehat{P}^\pi, o_h(\cdot) \sim \mathrm{Ber}(\widehat{F}_h(\cdot))} \left[ \sum_{h=1}^{H} \left[ (\widehat{\beta}_h(\ell_h) - \beta_h(\ell_h)) \cdot v \right] \cdot \mathbb{I}\{o_h(\pi(\ell_h)) = 1\} \right] \\
&\leq \mathop{\mathbb{E}}_{\mathcal{L} \sim \widehat{P}^\pi, o_h(\cdot) \sim \mathrm{Ber}(\widehat{F}_h(\cdot))} \left[ \sum_{h=1}^{H} \left| \widehat{\beta}_h(\ell_h) - \beta_h(\ell_h) \right| \cdot \mathbb{I}\{o_h(\pi_h(\ell_h)) = 1\} \right]
\end{aligned}
$$
(21)

$\square$

Second, we extend the well-known simulation lemma in standard RL [16] to our case. Please note both transition probability and expected utility function of the learner (Eq. (3)) depend on $F$.

**Lemma 3.** *For any fixed bidding strategy $\pi$, we have*

$$
\left| R(\pi; \theta, \widehat{F}^t) - R(\pi; \theta, F) \right| \leq (H^2 + 2H) \sqrt{\frac{\log(2H/\delta)}{2t}}.
$$
(22)

*holds with probability at least $1 - \delta$. In addition, we have with probability at least $1 - \delta$,*

$$
\mathrm{OPT}(\theta, F) - \mathrm{OPT}(\theta, \widehat{F}^t) \leq (H^2 + 2H) \sqrt{\frac{\log(2H/\delta)}{2t}}.
$$

*Proof.* Denote $\widehat{M}^t$ as the MDP induced by a bidding policy $\pi$ and parameters $\theta, \widehat{F}^t$. Similarly, let $M$ represent the MDP for a bidding policy $\pi$ with parameters $\theta$ and $F$. Let $\widehat{\Gamma}_h^t(\cdot|\ell, b)$ and $\Gamma_h(\cdot|\ell, b)$ be the transition at round $h$ for MDP $\widehat{M}^t$ and $M$, respectively.

Given the MDP re-formulation in Subsection 3.1, we have for any $\ell, b$ and $h = 2, 3, \cdots, H$,

$$
\begin{aligned}
\sum_{\ell'} \left| \widehat{\Gamma}_h(\ell'|\ell, b) - \widehat{\Gamma}_h^t(\ell'|\ell, b) \right| &= \left| \widehat{\Gamma}_h^t(\ell+1|\ell, b) - \Gamma_h(\ell+1|\ell, b) \right| + \left| \Gamma_h(1|\ell, b) - \widehat{\Gamma}_h(1|\ell, b) \right| \\
&= 2 \left| \widehat{F}_{h-1}^t(b) - F_{h-1}(b) \right| \leq 2 \sup_b \left| \widehat{F}_{h-1}^t(b) - F_{h-1}(b) \right|
\end{aligned}
$$

In addition, we denote $r_h$ as the expected reward function for the MDP $M$ at round $h$, similarly, we can also define $\widehat{r}_h^t$ for the MDP $\widehat{M}^t$. For any $(\ell, b)$, we can bound $|r_h(\ell, b) - \widehat{r}_h^t(\ell, b)|$ as below:

$$
\begin{aligned}
&\left| r_h(\ell, b) - \widehat{r}_h^t(\ell, b) \right| \\
&= \left| (\beta_h(\ell) \cdot v_h - \widehat{p}_h^t(b)) \widehat{F}_h^t(b) - (\beta_h(\ell) \cdot v_h - p_h(b)) F_h(b) \right| \\
&\leq \left| \beta_h(\ell) v_h (\widehat{F}_h^t(b) - F_h(b)) \right| + \left| b \widehat{F}_h^t(b) - b F_h(b) \right| + \left| \int_0^b \widehat{F}_h^t(v) dv - \int_0^b F_h(v) dv \right| \\
&\leq 3 \sup_b \left| \widehat{F}_h^t(b) - F_h(b) \right|,
\end{aligned}
$$

where the first equality is based on the definition of $r_h(\cdot, \cdot)$ and $\widehat{r}_h^t(\cdot, \cdot)$, the second inequality is based on triangle inequality and the last inequality holds because of the fact that $\beta_h(\ell) v_h \in [0, 1]$ and $b \in [0, 1]$.

By DKW inequality (Lemma 4) and union bound, with probability at least $1 - \delta$, we have for all $h \in [H]$

$$
\sup_b |\widehat{F}_h^t(b) - F_h(b)| \leq \sqrt{\frac{\log(2H/\delta)}{2t}}
$$

Given the above bound of transition and reward between two MDPs $M$ and $\widehat{M}^t$ and the Simulation Lemma (Lemma 6), with probability at least $1 - \delta$, we have

$$
\begin{aligned}
\left| R(\pi; \theta, \widehat{F}^t) - R(\pi; \theta, F) \right| &\leq H(H-1) \sqrt{\frac{\log(2H/\delta)}{2t}} + 3H \sqrt{\frac{\log(2H/\delta)}{2t}} \\
&= (H^2 + H) \sqrt{\frac{\log(2H/\delta)}{2t}}
\end{aligned}
$$
(23)

(24)

For notation simplicity, we denote $\pi^*$ as the optimal bidding policy corresponding to parameters $\theta$ and $F$, then we have

$$
\begin{aligned}
\mathrm{OPT}(\theta, F) - \mathrm{OPT}(\theta, \widehat{F}^t) &= R(\pi^*; \theta, F) - R(\pi^*; \theta, \widehat{F}^t) + R(\pi^*; \theta, \widehat{F}^t) - \mathrm{OPT}(\theta, \widehat{F}^t) \\
&\leq R(\pi^*; \theta, F) - R(\pi^*; \theta, \widehat{F}^t),
\end{aligned}
$$

where the first inequality holds because of the definition of $\mathrm{OPT}(\theta, \widehat{F}^t)$ and the second inequality is based on Eq. (22) for the fixed bidding policy $\pi^*$. Using the same argument as in Eq. (23), we complete the proof.

$\square$

Given the above two auxiliary lemmas, we are ready to prove Theorem 2.

*Proof of Theorem 2.* For notation simplicity, let

$$
\mathrm{OPT}(\widehat{\mathbf{CR}}^t, F) := \max_{\theta \in \widehat{\mathbf{CR}}^t} \mathrm{OPT}(\theta, F)
$$

In addition, we denote $\tau$ as the number of episodes for pure explorations. Note, in each pure exploration $t \leq \tau$, we exactly increase $n_h(\ell)$ by 1 for one $(h, \ell)$ pair. To make sure $n_h(\ell) \geq \frac{\log(4T/\delta)}{C_0}$, we only need $\tau \leq H^2 \frac{\log(4T/\delta)}{C_0}$.

We first propose a new decomposition of the expected regret $\mathrm{Regret}(T)$ in the following way,

$\mathrm{Regret}(T)$

$$
= T \cdot \mathrm{OPT}(\theta, F) - \mathbb{E}\left[\sum_{t=1}^{T} R(\pi^t; \theta, F)\right]
$$

$$
\leq H\tau + (T - \tau) \cdot \mathrm{OPT}(\theta, F) - \mathbb{E}\left[\sum_{t=\tau+1}^{T} R(\pi^t; \theta, F)\right]
$$

$$
= H\tau + \underbrace{(T - \tau) \cdot \mathrm{OPT}(\theta, F) - \sum_{t=\tau+1}^{T} \mathrm{OPT}(\theta, \widehat{F}^{t-1})}_{(i)} + \underbrace{\sum_{t=\tau+1}^{T} \mathrm{OPT}(\theta, \widehat{F}^{t-1}) - \sum_{t=\tau+1}^{T} \mathrm{OPT}(\widehat{\mathbf{CR}}^{t-1}, \widehat{F}^{t-1})}_{(ii)}
$$

$$
+ \underbrace{\sum_{t=\tau+1}^{T} \mathrm{OPT}(\widehat{\mathbf{CR}}^{t-1}, \widehat{F}^{t-1}) - \mathbb{E}\left[\sum_{t=\tau+1}^{T} R(\pi^t; \theta, \widehat{F}^{t-1})\right]}_{(iii)} + \underbrace{\mathbb{E}\left[\sum_{t=\tau+1}^{T} R(\pi^t; \theta, \widehat{F}^{t-1}) - \sum_{t=\tau+1}^{T} R(\pi^t; \theta, F)\right]}_{(iv)}
$$

$$
(25)
$$

The first inequality above is based on the fact the reward (expected utility of the learner) at each round is bounded by $[0, 1]$. Then we bound the above terms separately in the following.

**Term (i).** By Lemma 3, for any fixed $t \in [T]$, we have $|\mathrm{OPT}(\theta, F) - \mathrm{OPT}(\theta, \widehat{F}^t)| \leq (H^2 + 2H)\sqrt{\frac{\log(2H/\delta)}{2t}}$ holds with probability at least $1 - \delta$. Then by union bound over $t = \tau, \cdots, T - 1$, we have

$$
\begin{aligned}
(T - \tau) \cdot \mathrm{OPT}(\theta, F) - \sum_{t=\tau+1}^{T} \mathrm{OPT}(\theta, \widehat{F}^{t-1}) &= \sum_{t=\tau}^{T-1} \mathrm{OPT}(\theta, F) - \mathrm{OPT}(\theta, \widehat{F}^t) \\
&\leq (H^2 + H) \sum_{t=\tau}^{T-1} \sqrt{\frac{\log(2HT/\delta)}{2t}} \\
&\leq (H^2 + H)\sqrt{\log(2HT/\delta)T}
\end{aligned}
$$

holds with probability at least $1 - \delta$.

**Term (ii).** After pure explorations, for any $t \geq \tau + 1$, we have $4e^{-C_0 n_h(\ell)} < \delta/T$. Then by Theorem 1, we have with probability at least $1 - \delta/T$, the true parameter $\theta \in \widehat{\mathrm{CR}}^t$. Then by the definition of $\mathrm{OPT}(\widehat{\mathrm{CR}}^t, \widehat{F}^t)$, for any $t \in [T]$, we have $\mathrm{OPT}(\theta, \widehat{F}^t) - \mathrm{OPT}(\widehat{\mathrm{CR}}^t, \widehat{F}^t) \leq 0$ holds with probability $1 - \delta/T$. Taking union bound over $t = \tau, \cdots, T - 1$, then with probability at least $1 - \delta$, we have

$$\sum_{t=\tau+1}^{T} \mathrm{OPT}(\theta, \widehat{F}^{t-1}) - \sum_{t=\tau+1}^{T} \mathrm{OPT}(\widehat{\mathrm{CR}}^{t-1}, \widehat{F}^{t-1}) \leq 0.$$

**Term (iii).** For notation simplicity, let $\widetilde{\theta}^t \in \widehat{\mathrm{CR}}^t$ be the parameter corresponding to $\pi^{t+1}$. Actually, it is the Then we can bound Term (iii) as below,

$$\sum_{t=\tau+1}^{T} \mathrm{OPT}(\widehat{\mathrm{CR}}^{t-1}, \widehat{F}^{t-1}) - \sum_{t=\tau+1}^{T} R(\pi^t; \theta, \widehat{F}^{t-1})$$

$$= \sum_{t=\tau+1}^{T} R(\pi^t; \widetilde{\theta}^{t-1}, \widehat{F}^{t-1}) - \sum_{t=\tau+1}^{T} R(\pi^t; \theta, \widehat{F}^{t-1}) \tag{26}$$

$$\leq \sum_{t=\tau+1}^{T} \mathop{\mathbb{E}}_{(\ell_1^t, \cdots, \ell_H^t) \sim \widehat{P}^{\pi^t}, o_h(\cdot) \sim \mathrm{Ber}(\widehat{F}_h^{t-1}(\cdot))} \left[ \sum_{h=1}^{H} \left| \widetilde{\beta}_h^{t-1}(\ell_h^t) - \beta_h(\ell_h^t) \right| \cdot \mathbb{I}\left\{ o_h\left(\pi^t(\ell_h^t)\right) = 1 \right\} \right],$$

where the inequality is based on Lemma 2 in Appendix. Here we slightly abuse the notation, let $o_h^t := o_h\left(\pi^t(\ell_h^t)\right)$. An important observation is that $o_h^t$ can be viewed as the action the learner takes at round $h$ in the $t^{\text{th}}$ episode and it only contains two different options, i.e., 0 (lose) or 1 (win). This rewriting can help us reduce the dependency of regret bound with number of bids, and we can easily extend to handle continuous bid space. Let $n_h^t(\ell, o)$ represent the total number observations of pair $(\ell, o)$ up to round $h$ and $t^{\text{th}}$ episode. Then we have,

$$\text{Equation (26)} \leq \mathbb{E}\left[ \sum_{t=1}^{T} \sum_{h=1}^{H} C_2 \sqrt{\frac{\log(T/\delta)}{n_h^t(\ell_h^t, o_h^t)}} \cdot \mathbb{I}\{o_h^t = 1\} \right]$$

$$\leq C_2 \sqrt{\log(T/\delta)} \, \mathbb{E}\left[ \sum_{t=\tau+1}^{T} \sum_{h=1}^{H} \sqrt{\frac{1}{n_h^t(\ell_h^t, o_h^t)}} \cdot \mathbb{I}\{o_h^t = 1\} \right] \tag{27}$$

$$\leq C_2 \sqrt{\log(T/\delta)} \, \mathbb{E}\left[ \sum_{h=1}^{H} \sum_{\ell=1}^{H} \sum_{i=1}^{n_h^t(\ell,1)} \frac{1}{\sqrt{i}} \right]$$

$$\leq C_2 \sqrt{\log(T/\delta) H^3 T},$$

where the first inequality is based on the fact that $\widetilde{\theta}^t \in \widehat{\mathrm{CR}}^t$, $\forall t \in [T]$ and $n_h^t(\ell, 1) = n_h(\ell)$ in episode $t$ given the definition in Algorithm 2. The third inequality is based on Lemma 5.

**Term (iv).** Again, applying Lemma 3 and union bound over $t = 1, \cdots, T - 1$, we can bound term (iv) in the following way,

$$\mathbb{E}\left[ \sum_{t=\tau+1}^{T} R(\pi^t; \theta, \widehat{F}^{t-1}) - \sum_{t=\tau+1}^{T} R(\pi^t; \theta, F) \right] \leq \mathbb{E}\left[ \sum_{t=\tau+1}^{T} R(\pi^t; \theta, \widehat{F}^{t-1}) - R(\pi^t; \theta, F) \right]$$

$$\leq (H^2 + H) \sqrt{\log(2HT/\delta) T}$$

Putting it all together, we can bound the regret as below,

$$\mathrm{Regret}(T) \leq O\left( H^2 \sqrt{\log(HT/\delta) T} + H^2 \log(T/\delta) \right)$$

holds with probability at least $1 - \delta$. $\qquad \square$

## D   Auxiliary Technical Lemmas

In this section, we enumerate several useful technical lemmas used in this paper. First, we introduce the well-known Dvoretzky–Kiefer–Wolfowitz (DKW) inequality which is used to bound the gap between $\widehat{F}^t(\cdot)$ and $F(\cdot)$.

**Lemma 4** (DKW Inequality). *For any episode $t$, at each round $h \in [F]$, $\sup_b |\widehat{F}_h^t(b) - F_h(b)| \leq \sqrt{\frac{\log(2/\delta)}{2t}}$ holds with probability at least $1 - \delta$.*

Second, we describe the celebrated simulation lemma, which was introduced and named in [16]. For completeness of the presentation, we provide the proof for this lemma.

**Lemma 5** (Lemma 7.5 in [1]). *Consider arbitrary $T$ sequence of trajectories, $\{\ell_h^t, o_h^t\}_{h=1}^H$ for $t = 1, \cdots, T$, we have*

$$\sum_{t=1}^T \sum_{h=1}^H \frac{1}{\sqrt{n_t(\ell_h^t, o_h^t)}} \cdot \mathbb{I}\{o_h^t = 1\} \leq \sum_{h=1}^H \sum_{\ell=1}^H \sum_{i=1}^{n_h^t(\ell,1)} \frac{1}{\sqrt{i}}$$

Finally, we describe the well-known simulation lemma. For completeness of exposition, we provide a simple proof for this lemma.

**Lemma 6** (Simulation Lemma [16]). *Consider two different MDPs $M$ and $M'$ with the same state and action spaces, $\mathcal{S}$ and $\mathcal{A}$. If the transition ($P$ and $P'$ resp.) and reward functions ($r$ and $r'$ resp., bounded by $[0, 1]$) of these two MDPs satisfy*

$$\forall s \in \mathcal{S}, a \in \mathcal{A}, \sum_{s' \in \mathcal{S}} \left| P(s'|s,a) - P'(s'|s,a) \right| \leq \varepsilon_1, \text{ and } \forall h \in [H], \left| r_h(s,a) - r_h' \right| \leq \varepsilon_2,$$

*Then for every non-stationary policy $\pi$ and fixed initial state $s_1$, the two MDPs satisfy*

$$\left| \mathop{\mathbb{E}}_{\{s_h,a_h\}_{h=1}^H \sim M, \pi} \left[ \sum_{h \in [H]} r_h(s_h, a_h) \right] - \mathop{\mathbb{E}}_{\{s_h,a_h\}_{h=1}^H \sim M', \pi} \left[ \sum_{h \in [H]} r_h'(s_h, a_h) \right] \right| \leq \frac{H(H-1)}{2}\varepsilon_1 + H\varepsilon_2$$

$$(28)$$

*Proof.* First, we denote $V_h^\pi(s, M)$ as the total reward from round $h$ to $H$ when the state at round $h$ is $s$. Similarly we can define $V_h^\pi(s, M')$.

Then Eq. (28) is equivalent to show $|V_1^\pi(s_1, M) - V_1^\pi(s_1, M')| \leq \frac{H(H-1)}{2}\varepsilon_1 + H\varepsilon_2$ and we will prove it using the inductive hypothesis that for any state $s$, policy $\pi$ and round $h = 1, 2, \cdots, H$,

$$|V_h^\pi(s, M) - V_h^\pi(s, M')| \leq \frac{(H-h)(H-h+1)}{2}\varepsilon_1 + (H-h+1)\varepsilon_2$$

For $h = H$, the hypothesis is clearly true since reward of two MDPs at round $H$ only differs at most $\varepsilon_2$ and this establishes the base case.

For the inductive step, we utilize the recursive formula of $V_h^\pi(s, M)$ and $V_h^\pi(s, M')$ in the following way,

$$
\begin{aligned}
&\left| V_h^\pi(s, M) - V_h^\pi(s, M') \right| \\
=\ & \left| \mathop{\mathbb{E}}_{s' \sim P(\cdot | s,a), a=\pi(s)} \left[ r(s,a) + V_{h+1}^\pi(s', M) \right] - \mathop{\mathbb{E}}_{s' \sim P'(\cdot | s,a), a=\pi(s)} \left[ r'(s,a) + V_{h+1}^\pi(s', M') \right] \right| \\
\leq\ & \varepsilon_2 + \left| \sum_{s'} P(s'|s,a) V_{h+1}^\pi(s', M) - \sum_{s'} P'(s'|s,a) V_{h+1}^\pi(s', M') \right| \\
\leq\ & \varepsilon_2 + \left| \sum_{s'} P(s'|s,a) V_{h+1}^\pi(s', M) - \sum_{s'} P'(s'|s,a) V_{h+1}^\pi(s', M) \right| \\
& + \left| \sum_{s'} P'(s'|s,a) \cdot \left( V_{h+1}^\pi(s', M) - V_{h+1}^\pi(s', M') \right) \right| \\
\leq\ & \varepsilon_2 + (H-h) \cdot \sum_{s'} \left| P(s'|s,a) - P'(s'|s,a) \right| + \sum_{s'} P'(s'|s,a) \left| V_{h+1}^\pi(s', M) - V_{h+1}^\pi(s', M') \right| \\
\leq\ & \varepsilon_2 + (H-h)\varepsilon_1 + \frac{(H-h-1)(H-h)}{2} \varepsilon_1 + (H-h)\varepsilon_2 \\
=\ & \frac{(H-h)(H-h+1)}{2} \varepsilon_1 + (H-h+1)\varepsilon_2,
\end{aligned}
$$

where the first inequality holds since $|r(s,a) - r'(s,a)| \leq \varepsilon_2$; the second inequality is based on the triangle inequality; the third inequality is because of the fact that $V_{h+1}^\pi(s, M) \leq H - h$ (recall the reward at each round is bounded by $[0, 1]$); the fourth inequality holds due to the inductive hypothesis. Then we complete the proof.

$\square$