# OpenReview forum: "Incrementality Bidding via Reinforcement Learning under Mixed and Delayed Rewards"
_NeurIPS.cc/2022/Conference — NeurIPS 2022 Accept_

### Official Review · Reviewer_Bj5w · 2022-07-10

**Rating:** 7
**Confidence:** 3
**Soundness:** 3 good
**Presentation:** 4 excellent
**Contribution:** 3 good

**Summary:**

The authors study a practically-motivated learning-to-bid problem in the online setting, motivated by the importance of incrementality bidding. To capture the causal effects of incrementality, the authors pose and analyze an MDP formulation, while past work operated under simplifying assumptions that yielded bandit formulations. Combining the MDP formulation with a novel parameter estimation algorithm, the authors establish regret bounds on the overall procedure.

**Questions:**

As far as I can tell the work is theoretically sound and is a good contribution. I’m not intimately familiar nor up-to-date on the literature on bidding/autobidding/what is the state of the art with respect to these things in practice. Thus, when reading the paper, some of the important terms like “conversion”, “incrementality bidding”, and so on were unfamiliar to me and I had to do some digging to get some understanding. If somehow the bidding landscape could be set up in a more reader-friendly way, that would be good. But, I think for someone who is more familiar with this field the paper would be accessible and well written.

Are there experiments that the authors could include that would help the theory? I understand that this is a theory paper that complements Lewis and Wong, but perhaps an experimental comparison to the results of that paper would be interesting.

**Strengths And Weaknesses:**

This work seems to fill in an important gap in the theoretical learning-to-bid literature. It relaxes assumptions that allowed previous work to study a simplified bandit formulation, and provides theoretical guarantees for an important practical setting introduced by Lewis and Wong.

---

> ### Author Response · Authors · 2022-08-02
> **To Reviewer Bj5w**
>
> Thanks for your helpful feedback! We will definitely add more high-level descriptions of these notations in the future version of the paper. Thanks for your suggestion: we agree adding experiments (especially real-world studies) will be an interesting future direction.

---

### Official Review · Reviewer_hB3J · 2022-07-11

**Rating:** 7
**Confidence:** 4
**Soundness:** 3 good
**Presentation:** 4 excellent
**Contribution:** 4 excellent

**Summary:**

This work considers an online ad-bidding problem where the reward can be mixed and delayed. It models the problem using an MDP,  and it proposes an UCB-based RL algorithm to solve this problem with an analysis of the regret, with a moment matching routine for parameter estimation that addresses the mixed and delayed reward issue.

**Questions:**

This is a very interesting work, some of the following may be suggestions for future consideration, which may or may not be necessary for the current version.

- In the paper, \beta_h(l) is assumed to be lower bounded by some positive number c_{\beta} to indicate that any previous ad presence can only increase the conversion rate. However, in practice, sometimes this can be negative as well, for example seeing too many ads that one is not interested in can have a negative impact on certain users. Would the current framework be able to accommodate the settings where  \beta_h(l) is negative as well?

- The MDP considered currently has states related to how long time ago one user has seen this ad before. Would some other related features be helpful for the modeling too? In that case, the states will be more complicated, would the current solutions still apply, or can be easily modified?

- What is the high-level idea and intuition of this moment matching algorithm? It will be helpful to explain the algorithm and intuition more clearly in the main body of the paper.

- It would be great if some synthetic or even real-life experiments can be presented to further illustrate the power of this method.


**Ethics Review Area:**

["I don’t know"]

**Limitations:**

Not applicable.

**Strengths And Weaknesses:**

Strength: This is a very interesting work considering mixed and delayed rewards in an online setting, and it proposes a method to solve these challenges.

Weakness: certain assumptions of the model may be a bit strong (please see the next section).

---

> ### Author Response · Authors · 2022-08-02
> **To Reviewer hB3J**
>
> Thanks for your helpful feedback! We answer your questions as below:
>
> 1. This is a good question. To respect the rebuttal policy, we don’t want to claim new results that were not included in the original paper. However, our results still hold for negative $\beta_h(\ell)$ as long as $|\beta_h(\ell)| > c$ for some positive constant $c$ and the time-varying rate $r(\tau)$ is non-negative. We will add a brief discussion on this in the final version.
> 2. It is possible to have more features in the state and we believe our analysis will still go through as long as the number of states is still finite. However, as you mentioned, the states will be more complicated and the regret bound will depend on the number of states (similar to standard RL).  We will also add this discussion in the final version of the paper.
> 3. We provided the high-level idea and intuition of the PAMM algorithm on the top (in our response to all reviewers). Hope this can answer your question, and we will add it in the main body of the paper.
> 4. Thanks for your suggestion: indeed, adding experiments (especially real-world studies) will be an interesting future direction.

---

> > ### Comment · Reviewer_hB3J · 2022-08-07
> > **Thanks for the response**
> >
> > Thanks for the detailed response! I believe this is interesting work and will keep my score.

---

### Official Review · Reviewer_ia2P · 2022-07-11

**Rating:** 7
**Confidence:** 3
**Soundness:** 4 excellent
**Presentation:** 4 excellent
**Contribution:** 3 good

**Summary:**

This paper proposes a new problem setting in reinforcement learning which is specialized for modeling incrementality bidding. Authors claim that this is a more realistic model compared with previous learning-to-bid models. The main difficulty in solving this RL problem is the mixed and delayed feedback, which is tackled by a novel pairwise moment-matching algorithm. The final regret bound is $\widetilde{O}(H^2\sqrt{T})$, which does not depends on the number of actions due to the special structure of bidding.

**Questions:**

- Can authors provide some examples that satisfy auctions happening every unit time?


**Limitations:**

I am not aware of any potential negative societal impact

**Strengths And Weaknesses:**

First of all, I need to mention that I am not very familiar with (learning to) bidding literature so there might be some misunderstanding.

Strengths:
- The MDP model is more realistic than previous learning-to-bid models because it considers multiple bidding for one user and the delayed and mixed feedback while the authors claim that previous models only consider single bid and thus use contextual bandits as the model.
- A new parameter estimation algorithm for delayed and mixed feedback.
- The paper is well written and easy to follow.

Weakness:
- The model still makes some strong assumptions. First, it is assumed that auctions happen per unit time. Second, it is assumed that the dependence of incrementality function on previous actions is only on the elapsed time since the last bid was won, but I understand that this may be necessary for theoretical analysis and it does make sense to me.
- The related work part is mainly on bidding literature and there should be more discussion on RL literature, e.g., a detailed comparison of previous works on delayed feedback in RL.

---

> ### Author Response · Authors · 2022-08-02
> **To Reviewer ia2P**
>
> Thanks for your helpful feedback! We will add a more detailed comparison with previous work on delayed feedback in RL. Beyond delayed feedback, the reward feedback in this paper is also mixed.
>
> This assumption is indeed for theoretical analysis, however we believe our algorithm would work reasonably well even though we allow auctions to happen in a continuous time. For example, we can always discretize the continuous time space to a finite number of intervals and our results still go through if the incrementality only depends on the time gap of the discrete intervals. On the other hand, in the real ad auction system, it is practical to assume auctions happen in unit time (e.g. millisecond) since there are billions of auctions (even for users with similar profiles) happening per day.

---

> > ### Comment · Reviewer_ia2P · 2022-08-06
> > **Thanks for your answer**
> >
> > Thanks for your answer! After reading other reviews and replies, I still believe this is an interesting contribution to the community and would like to keep my score.

---

### Official Review · Reviewer_ZwJW · 2022-07-13

**Rating:** 6
**Confidence:** 2
**Soundness:** 3 good
**Presentation:** 2 fair
**Contribution:** 2 fair

**Summary:**

This paper first uses the MDP to formulate the sequential bidding process, and then studies the problem of optimizing the a specific sequence with unknown incrementality parameters of the bidding process.

The incrementality rate function $d_h$ is defined to enjoy the Markovian property in Assumption 1 and plays the role of "reward" in the MDP. The state $\ell$ is formulated by the number of rounds between the last winning of the learner and the current round. Therefore, the transition function is defined by the distribution of the highest bid among other bidders $F_h(\cdot)$. A special problem is that the incrementality has to be calcuated with Poisson process estimation.

The algorithm is designed based on the principle "Optimism on the face of uncertainty" and the concentration inequalities of the parameters of the MDP. For each episode, it selects the optimal policy associated with the parameters $(\beta,\lambda)$ within the confidence sets, and adjust the confidence interval accordingly. It achieves similar regret bound $O(\sqrt{T})$ without knowing the environments beforehand.

**Questions:**

By the work of Simchowitz and Jameison (2019) we know that it is possible to achieve $O(\log T)$ regret for unknown transition MDP problems. Is it able to achieve some problem-dependent / gap-dependent regret bounds ?

**Limitations:**

None.

**Strengths And Weaknesses:**

I am willing to reconsider my scores if the authors could provide a more detailed argument or some references for the following concerns.

1. The idea of proposed algorithm is very similar to that of the UCRL2 or UCBVI and the concentration inequalities are the same, which makes the technical contribution of this work less obvious.

2. It would be better to emphasize the procedure to estimate the incrementality. It is hard to understand the usage of $X$ and $Y$ in the PAMM algorithm and the necessarity.

There are many notations in this paper and some of them are not clearly defined before using, such as $n_t$ and $k_t$ in lines 293,295.

---

> ### Author Response · Authors · 2022-08-02
> **To Reviewer ZwJW**
>
> We appreciate the reviewer’s positive feedback and willingness to improve the score after our clarification. We now address the reviewer’s two concerns as follows:
>
> ``The idea of the proposed algorithm is similar to UCRL2 or UCBVI and the concentration inequalities are the same, which makes the technical contribution of this work less obvious."
>
> A: We would like to emphasize two key novelties in our algorithm design:
> - Our regret bound does not depend on the number of actions (i.e., number of possible bids in our model), whereas the regret of both UCRL2 and UCBVI depends on the number of agent actions (which is infinite in our problem). Our better regret bound is due to the special structure of our problem (see line 85-86), and thus requires different ways to choose the action/bid at each round and also different analysis of the algorithm. We do acknowledge that our algorithm is based on the optimism in face of uncertainty, which is very widely used in RL research, just like UCRL2 and UCBVI (this is probably why the reviewer feels that the algorithm sketch appears “similar”).
> - Another key novelty is our PAMM algorithm, designed specifically for estimating the parameter of the Poisson process. This algorithm is new, and may be of independent interest for future research on RL with mixed rewards. Note that standard parameter estimation approaches in most previous RL algorithms — e.g., simply using the empirical mean or linear regression for linear reward setups — are not applicable to our problem with mixed reward. Moreover, the classic maximum likelihood estimation does not have desirable guarantees due to lack of convexity in our problem.  Therefore, it is not at all a trivial task to come up with a provably correct parameter estimation method under mixed reward feedback.
>
> Finally, we note that the concentration rate $1/\sqrt{n_t}$ of our moment-matching estimation algorithm is indeed similar to the concentration rate of simply the empirical mean or linear regression. However, such a $1/\sqrt{n_t}$ concentration rate is a fundamental phenomenon in statistics – to be specific, they all originate from the central limit theorem. The difficulty here is to come up with the right estimation method to achieve this concentrate rate (once there is a method, the best rate typically is $1/\sqrt{n_t}$).
>
> ``Emphasize the procedure to estimate the incrementality. It is hard to understand the usage of X and Y in the PAMM algorithm and the necessity."
>
> A: Thanks for the thoughtful suggestion. We provide a high level idea and intuition of the PAMM algorithm on the top (in our response to all reviewers), and will add more intuitions about the algorithm in the future version.  The definition of X and Y are defined in Eq. (7) and they model the difference of the number of the conversions between two episodes. We will clarify it in the future version.
>
> Regarding notations, $n_t = k_t$ and they denote the number of winning events (showing the ad) in the episode $t$. We will go over the paper and make it clear in the final version.
>
> For your question regarding instance-dependent regret:
>
> A: If there exists a reward gap among different actions, then indeed we believe it is also possible to achieve O(log T) regret. However, this may not be realistic in our problem since the bids (i.e., agent actions) can be continuous. This is why we went with instance-independent regret.

---

> > ### Author Response · Authors · 2022-08-10
> > **Reminder**
> >
> > Dear Reviewer ZwJW,
> >
> > Since today is the final day for author-reviewer discussion period and we haven't heard back from you, please let us know if our response makes sense to you and we are happy to answer your questions at the last time.

---

### Author Response · Authors · 2022-08-02
**To all reviewers**

Since several reviewers asked about the intuition and high-level idea of the PAMM algorithm, in this global response, we provide a high-level description on PAMM here.

The PAMM algorithm is specifically designed for estimating parameters from mixed reward signals, each drawn from a Poisson process. Its primary advantage lies in its online nature, guaranteed convergence and efficient computation. This is achieved by the following observations. First, all information about reward scale parameter $\beta_h$ and reward variance parameter $\lambda_h$ is intrinsically reflected only in the “differential behaviors” for those episodes which had conversions vs. those episodes which did not have conversions at time $h$, conditioning their matched history before time $h$ as defined in the sample matching step. This is due to the Markovian property for the Poisson process, and our moment matching is hinged into this property. Second, for the Poisson distribution (as a member of the exponential family), the method of matching the canonical parameter (mean) gives an equivalent estimator as the MLE in a single Poisson case (but not necessary in our more complicated model). These two ideas motivate our procedure, by matching the episodes, we precisely locate the needed Poisson signals and solve the estimators by the mean value structure. Because of this, the PAMM can recover the parameters with essentially optimal rate (in the online sense), even though we do not use the full likelihood that would result in non-convexity and an unpleasant gap between the theory and the practical algorithm.

---

### Meta-Review · Area_Chair_edv7 · 2022-08-26

**Recommendation:** Accept
**Confidence:** Certain

**Metareview:**

The paper studies sequential bidding problems by formulating them as MDPs (previous attempts made simplifying assumptions that yielded bandit formulations). The reviewers found the arguments about the causal effects of incrementality requiring richer modeling very convincing, and appreciated the difficulty of the problem due to delayed and mixed rewards. This necessitated a new combination of pairwise moment matching with optimism in the face of uncertainty.

The authors clarified the reveiwers' technical questions about the PAMM algorithm during the feedback phase, and reviewers reached consensus that the paper's contributions are novel, interesting, and likely to lead to future research and real-world applications.


**Award:**

No

---

### Decision · Program_Chairs · 2022-09-14

Accept